# Stress, Anxiety, and Depression for Chinese Residents in Japan during the COVID-19 Pandemic

**DOI:** 10.3390/ijerph18094958

**Published:** 2021-05-07

**Authors:** Qin Hu, Maki Umeda

**Affiliations:** 1Graduate School of Nursing Art and Science, University of Hyogo, Akashi 673-8588, Hyogo, Japan; 2Research Institute of Nursing Care for People and Community, University of Hyogo, Akashi 673-8588, Hyogo, Japan; maki_umeda@cnas.u-hyogo.ac.jp

**Keywords:** COVID-19, foreign residents, mental health, disaster preparedness, social support

## Abstract

The coronavirus disease 2019 (COVID-19) has profoundly affected the psychological well-being of foreign residents. This study examines stress, anxiety, and depression levels in Chinese residents in Japan during the COVID-19 pandemic. It identifies risk factors and the roles of disaster preparedness and social support. An online survey among Chinese residents in Japan was conducted from 22 June to 14 July 2020. The Depression, Anxiety and Stress Scale, Disaster Preparedness for Resilience Checklist, and Social Support Rate Scale were used to measure psychological symptoms. Multivariable linear regressions identified the risk factors and positive effects of disaster preparedness and social support. Of the total 497 participants, 45.3%, 66.6%, and 54.3% reported severe stress, anxiety, and depressive symptoms, respectively. People with a lower level of education, a higher level of economic influence, the presence of COVID-19 symptoms, and confirmed or suspected family or friends in China were associated with higher levels of stress, anxiety, and depression. This study, to the best of our knowledge, is the first survey to reveal the protective role of disaster preparedness in reducing psychological symptoms during the pandemic. It offers unique data for further research on how to promote the mental health of vulnerable populations including foreign residents.

## 1. Introduction

Coronavirus disease 2019 (COVID-19) causes a severe acute respiratory disease that is considered equivalent to severe acute respiratory syndrome (SARS) and the Middle East respiratory syndrome (MERS) albeit with a completely different clade [1]. As of 12 January 2021, over 88 million cases and 1.9 million deaths were reported to the World Health Organization (WHO) [2].

Previous research has shown that an international pandemic has negative psychological impacts on those affected [3,4,5]. In particular, the existing literature has emphasized the high prevalence of stress, anxiety, and depression during SARS and MERS [6,7]. Similar symptoms occurred during the COVID-19 pandemic all over the world. For example, the statistics revealed that, among the public citizens of China, 35.1% had anxiety, 20.1% had depression, and 8.1% experienced stress [8,9,10]; similarly, 21.6% of the Spanish population had anxiety and 18.7% had depression [11]. In Australia, where the pandemic was under control [12], 7.7%, 11.7%, and 9.6% of the adults were found to experience anxiety, depression, and stress, respectively [13].

After the onset of COVID-19, foreign residents who were vulnerable during COVID-19 were prone to mental health consequences, as in other types of disasters [14]. Their vulnerabilities stemmed from difficulties with limited information gathering because of language gaps and insufficient direct emotional support from close relatives [15,16]. In addition, the research has assumed that foreign residents whose country of origin was severely affected by COVID-19 might face heavier psychological consequences [17]. China experienced the first outbreak of COVID-19 without any provable efficient treatment and had to explore ways to control the pandemic [18]. As COVID-19 was first reported in China, many Chinese residents in foreign countries experienced stigma and discrimination [19], and others experienced psychological symptoms. Over 40% of Asian-Americans reported an increase in anxiety and depressive symptoms [20]. To date, studies examining the mental condition of Chinese individuals living abroad remain limited.

Japan, as China’s neighbor, shares a long history of trade and cultural exchange. In 2020, nearly 3 million foreign residents lived in Japan, and 27% of them were Chinese, which makes the Chinese the largest group of foreign residents in Japan [21]. Over 36% of the Chinese residents in Japan reportedly experienced a significant economic impact from the pandemic [15], leading to difficulties from job loss or revenue deduction [22]. Moreover, social stigma targeted Chinese individuals, which adversely affected their mental health further [23]. Since the COVID-19 outbreak became widely known from its incidence in Wuhan, China [24], Chinese individuals became at a higher risk for being targeted by violence and stigma [25]. In Japan, concerns regarding a tendency of exclusionary attitude toward foreign residents, particularly Chinese residents, were addressed under the threat of infection [26].

Disaster preparedness helps people respond to disasters and saves lives. It reduces the negative impact of disasters and prevents related mental health consequences [27,28]. Fear of an uncertain future disaster imposes constant pressure and psychological distress, as it has in the case of COVID-19 [29]. Disaster preparedness helps people feel prepared, improves their sense of security, and allows them to identify their feelings and manage their emotional responses to better cope with difficulties, thus reducing anxiety and depression [30,31]. Therefore, when individuals prepare for material and psychological aspects, fewer psychological symptoms may occur [32].

COVID-19 prevention strategies, such as social distancing and home quarantine, challenged conventional means of providing and receiving social support and highlighted their importance in preventing mental problems [33]. The physical and social isolation caused by social distancing may cause individuals to lose their emotional ties with significant others, thereby leading to weaker social support [34]. Social support decreased anxiety levels during COVID-19 [35]. However, those living in foreign countries were likely living alone as well as at a considerable distance from their relatives and acquaintances, which heightened the likelihood of their experiencing psychological symptoms.

This study examined the prevalence and associated risk factors of stress, anxiety, and depression among Chinese residents in Japan during the COVID-19 pandemic. It also explored the protective effects of disaster preparedness and social support on mental health in the context of the pandemic. It is hoped that the results of the present research will help the supporting agencies of foreign residents as well as medical staff protect and promote the mental health of foreign residents during the COVID-19 pandemic in Japan and possibly in other countries.

## 2. Materials and Methods

### 2.1. Study Design and Participants

In this study, a cross-sectional survey was performed with a sample of Chinese nationals who lived in Japan. Data were collected from 22 June to 14 July 2020. The questionnaires were distributed via an online survey system (Wenjuanxing) alongside an appended consent form through a primary online communication means, such as WeChat, Weibo, and QQ. All of the aforementioned platforms are well-developed communication systems in China that gather people with the same purpose or interests into groups. In addition, the individuals in these groups were encouraged to send the link to the survey to their social circles. The inclusion criteria for the participants were as follows: (1) people whose country of origin was China; (2) who were aged 18 years old or above, regardless of their immigration status or the duration of their stay in Japan; and (3) who had lived in Japan from December 2019 to 14 July 2020.

### 2.2. Ethical Considerations

All the participants responded to the anonymous survey on a voluntarily basis; informed consent was obtained from the participants. Personal information regarding the participants was not collected to ensure data confidentiality. The instructions provided were explicitly stated, and the participants could interrupt or withdraw from the survey at any time with ease.

### 2.3. Measures

The self-reported 21-item Depression Anxiety Stress Scale (DASS-21) is an instrument that measures an individual’s mental status in relation to depression, anxiety, and stress [36]. The scale has seven items for each subscale of depression, anxiety, and stress, with four choices ranging from 0 (did not apply to me at all) to 3 (very much applied to me). The total scores of stress are divided into the following categories: normal (0–7), mild (8–9), moderate (10–12), severe (13–16), and extremely severe (over 17). Similarly, the total scores of anxiety are classified as normal (0–3), mild (4–5), moderate (6–7), severe (8–9), and extremely severe (over 10). Finally, the total scores of depression are divided as follows: normal (0–4), mild (5–6), moderate (7–10), severe (11–13), and extremely severe (over 14) [36]. The Chinese version of the DASS-21 has a Cronbach’s α of 0.89 [37].

The Disaster Preparedness for Resilience Checklist (DPRC) was constructed based on the Inter-Agency Standing Committee’s guidelines for mental health and psychosocial support in emergency settings. The DPRC measures basic service, community, and family support as well as psychological preparation. It was developed organically and has been validated in Japanese [32]. The DPRC comprises 23 items concerning two domains: disaster preparedness and psychological preparedness, and has a Cronbach’s α of 0.83 [32].

The Social Support Rate Scale (SSRS), which was created by Xiao [38], measures the type and levels of social support received during the pandemic. The SSRS includes 10 items that measure three types of social support: Subjective Support (four items), Objective Support (three items), and Support Availability (three items). The scale ranges from 12 to 65, with higher scores indicating higher levels of social support. The Cronbach’s alpha for internal consistency for the use of the SSRS was 0.808 in a previous study [39].

The covariates herein were demographic variables, pandemic-related variables, and factors related to a Chinese person living in Japan, which is associated with deteriorated mental health during the pandemic [8,9,10]. The demographics were measured by age (18–29 = 1, 30–39 = 2, or 40 and over = 3); sex (male = 1 or female = 2); educational level (high school or below = 1 or college and above = 2); marriage (married = 1 or unmarried = 2); and level of economic influence (no influence = 1, less than 50% of monthly income = 2, or more than 50% of monthly income = 3). Three dummy variables were created for occupational categories: medical staff, educator, and others (i.e., student). COVID-19 related variables were a participant’s location in Japan (no-risk area = 1 or high-risk area = 2); the time they focused on news related to COVID-19 (under 1 h = 1, 1–2 h = 2, or more than 3 h = 3); whether they had had COVID-19 symptoms (no = 1 or yes = 2); and whether they had had a polymerase chain reaction (PCR) test (no = 1 or yes = 2). High-risk areas declared an emergency with a higher amount of people affected during the data collection period, including the Hokkaido, Kanto, and Kinki districts [40]. Factors related to the participants’ residence in Japan were measured in terms of whether the participant lived with Japanese individuals (yes = 1 or no = 2), their Japanese language level (good = 1 or so-so = 2), and whether they had confirmed or suspected family or friends in China (no = 1 or yes = 2).

### 2.4. Data Analysis

Descriptive statistics were calculated for the demographic characteristics as well as the variables related to COVID-19 and the participants’ residence in Japan; these were subsequently expressed as frequencies (%) and mean scores. The prevalence of stress, anxiety, and depression was defined with the criteria formulated by Antony et al. [36]. Regression coefficients were calculated through multiple linear regressions with each of the stress, anxiety, and depression scores of the DASS-21 as dependent variables. The total scores of the DPRC, SSRS, and all the covariates were entered into the models simultaneously to examine the independent association of disaster preparedness and social support with stress, anxiety, and depression after controlling for covariates. The significance level was set as *p* < 0.05. The statistical analysis was performed using SPSS Statistic 21.0 (IBM SPSS Statistics, Chicago, IL, USA).

## 3. Results

### 3.1. Participant Characteristics

In total, the number of Chinese participants who lived in Japan during the COVID-19 pandemic was 497. Almost 60.4% of the participants were between 30 and 39 years old; further, 51.5% of them were male. In addition, 66% of the participants were married. Approximately 63.2% experienced less than a 50% economic impact because of COVID-19 at the time of data collection. Most participants (55.3%) reported living in high-risk areas such as the Tokyo, Osaka, and Hokkaido prefectures, and 94.8% had been in Japan for 10 years or less. Of the participants, 69.8% had experienced no COVID-19 symptoms in the past 14 days, and 95.8% of all the participants had not taken a PCR test. Moreover, 68.2% of the entire sample had suspected family or friends in China, while 32.4% had confirmed family or friends in China. Table 1 presents further detail.

### 3.2. Descriptive Statistics for Stress, Anxiety, and Depression Using the DASS-21

The text continues here (Table 2). The outcomes of stress, anxiety, and depression, as measured by the DASS-21, showed that nearly or over half of the participants had extremely severe levels of stress (45.3%), anxiety (66.6%), and depression (54.3%) at the time of data collection (Table 2).

### 3.3. Association of Disaster Preparedness and Social Support with Depression, Anxiety, and Stress

Table 3 shows the results of multiple linear regression models between all the variables and the DASS-21. After controlling for other independent variables, disaster preparedness for resilience had significantly negative relations with stress (b = −0.09, *p* < 0.01), anxiety (b = −0.11, *p* < 0.01), and depression (b = −0.12, *p* < 0.01). Similarly, social support was also negatively related to stress (b = −0.12, *p* < 0.01), anxiety (b = −0.17, *p* < 0.01), and depression (b = −0.20, *p* < 0.01) independent of other risk factors. Among the risk factors, the participants with higher education levels were more likely to have lower levels of stress (b = −0.93, *p* < 0.05), anxiety (b = −0.80, *p* < 0.05), and depression (b = −0.43, *p* < 0.05) symptoms. Those who reported a higher economic influence level were more likely to have higher levels of stress (b = 0.83, *p* < 0.05), anxiety (b = 0.84, *p* < 0.05), and depression (b = 0.97, *p* < 0.05) symptoms. The participants with COVID-19 symptoms were more likely to have higher levels of stress (b = 2.44, *p* < 0.01), anxiety (b = 1.82, *p* < 0.01), and depression (b = 0.98, *p* < 0.05) symptoms. Those who had confirmed family or friends in China were more likely to have higher levels of stress (b = 1.11, *p* < 0.01), anxiety (b = 1.14, *p* < 0.01), and depression (b = 0.99, *p* < 0.05) symptoms. Likewise, the participants with suspected family or friends in China were more likely to have higher levels of stress (b = 1.30, *p* < 0.01), anxiety (b = 1.24, *p* < 0.01), and depression (b = 0.10, *p* < 0.05) symptoms.

## 4. Discussion

This online-based cross-sectional study provided evidence for the high prevalence of stress, anxiety, and depression among Chinese individuals living in Japan. The related risk factors were as follows: a lower level of education, a higher level of economic influence, the presence of COVID-19 symptoms, and confirmed or suspected family or friends in China. Furthermore, the protective effects of disaster preparedness and social support reduced psychological symptoms for Chinese residents in Japan during the pandemic.

The present research highlighted the deteriorated psychological state of Chinese individuals in Japan during the COVID-19 pandemic [23]. The results showed that 45.3% of the participants reported that they experienced stress during the pandemic, 66.6% reported heightened anxiety, and 54.3% reported experiencing depression. The prevalence of these symptoms in the current sample was comparatively higher than what was previously reported from a survey of 2000 local Japanese respondents, wherein a total of 10.9% expressed pandemic-related anxiety and 17.3% reported depressive symptoms [41]. Studies of foreign residents in other countries have shown similar results. For example, 20% of foreign workers in Italy indicated that they had experienced depression, and 23% reported symptoms of anxiety during COVID-19—rates that were higher than those found among Italy’s native citizens [17]. Thus, foreign residents, as a group, appear to be more vulnerable to increases in stress, anxiety, and depression in relation to COVID-19 than native-born citizens, a finding that agrees with previous reports [42,43]. These results indicate that governments must pay more attention to mental health problems among foreign residents during the pandemic.

The identification of risk factors is crucial for the early prevention and protection of vulnerable groups. The results herein suggest that individuals with lower levels of education tended to experience severe stress, anxiety, and depression symptoms during the pandemic, similar to a recent Italian study that indicated that such individuals were associated with higher levels of mental distress symptoms [44]. The participants’ economic level was also a risk factor in the present research, which aligns with previous studies of Japanese citizens during the pandemic [45]. For risk factors under COVID-19, the study found that COVID-19 entailed poor mental health, which aligns with earlier observations [9]. As for the cultural issues encountered by Chinese residents in Japan, unlike in an earlier study, whether native Chinese citizens had COVID-19 symptoms had nothing to do with their psychological distress [9]. This result can be explained by the extra psychological burden on foreign residents with respect to worrying about their relatives and friends.

This study shows the protective role of disaster preparedness for foreign residents during the COVID-19 pandemic. Previous studies primarily examined disaster preparedness with respect to mental health in the setting of natural disasters [46,47], whereas the current study examined the association with the background of the COVID-19 pandemic. Restrictive measures such as lockdowns and quarantines for those exposed to the infectious disease have revealed as some of the most efficient actions in terms of controlling the spread of COVID-19 [48]. This research indicates that disaster preparedness is extremely important to maintain individual mental health and to meet the basic needs of life in the event that a government announces a sudden lockdown to minimize the spread of a virus. The protective role of disaster preparedness for mental health might be because it helps individuals obtain a feeling of being well prepared, improves their sense of security, and increases the use of positive coping [30,31]. Therefore, interventions to promote disaster preparedness for vulnerable populations, including foreign residents, are a promising solution for minimizing the psychological impact of an unpredictable health emergency. Public health workers should consider collaborating with the communities of foreign residents or support organizations to provide multilingual disaster preparedness services.

Social support helped maintain Chinese mental health during the pandemic in Japan, which corroborates a previous Korean investigation on MERS [15] and an Asian-American study on COVID-19 [20]. During the COVID-19 pandemic, staying at home and avoiding social contact impaired the ability of many to obtain and maintain relations. Although conventional means of social support were hampered during the pandemic, our results emphasize the vital role of social support and indicate the need for alternative means of social connections for foreign residents, such as web platforms or smartphone applications.

## 5. Limitations

The participants were recruited through a convenience sampling method using the groups of the author’s on social networking services. This method might not have adequately reflected the Chinese population in Japan. Moreover, the questionnaire, which was conducted online to avoid possible infection during the pandemic, excluded non-internet users. Finally, the generalizability of the findings herein might be limited to Chinese individuals who lived in Japan during a specific period. Future studies among different population groups could confirm the applicability of our results. To understand how disaster preparedness can protect mental health during a pandemic, examining factors that explain beneficial conditions, such as better coping as well as an increased sense of security and self-efficacy, should be further investigated.

## 6. Conclusions

The current research found that the mental health of Chinese residents in Japan deteriorated during the COVID-19 pandemic. The present results highlight the fact that foreign residents, particularly those with families or friends affected by COVID-19 in their countries of origin, have a need for care that goes beyond that which is focused on citizens living in their native countries. Assisting remote communication between those affected and those living abroad may ease worry, improve mental health, and offer emotional support.

This is, to the best of the authors’ knowledge, the first study that connects the protective role of disaster preparedness with mental health during a pandemic. The generalizability of the findings herein must be tested with individuals from other cultural backgrounds using a longitudinal study design. In addition, social support was another protective factor that reduced stress, anxiety, and depression symptoms during the pandemic. Since we live in a socially distanced world under an ongoing pandemic, foreign residents must have a platform for mutual social support that accommodates their language and cultural needs, possibly through the use of information technology. This study provides important data that can be used to engender interventions and future studies that aim at protecting and promoting the mental health of foreign residents during health emergencies.

## Figures and Tables

**Table 1 ijerph-18-04958-t001:** Participant characteristics (*n* = 497).

	*n* = 497	
Demographic Variables	*n* (Mean)	%
Age (mean ± SD)	33.7 ± 6.22	
18–29	112	22.5
30–39	300	60.4
40 and over	85	17.1
Sex		
Male	256	51.5
Female	241	48.5
Education level		
High school and below	175	35.2
College and above	322	64.8
Occupation		
Student	82	16.5
Medical staff	52	10.5
Educator	42	8.5
Other	321	64.5
Marriage		
Married	328	66
Unmarried	169	34
Level of economic influence		
No influence	61	12.3
Less than 50%	314	63.2
More than 50%	122	24.5
**Variables related to COVID-19**		
Location in Japan		
Not a high-risk area	275	55.3
High-risk area	222	44.7
Time focused on COVID-19 news		
Less than 1 h	194	39
1–2 h	198	39.8
More than 3 h	105	21.1
Had COVID-19 symptoms		
No	347	69.8
Yes	150	30.2
Had a PCR test		
Yes	21	4.2
No	476	95.8
**Variables related to Chinese residents living in Japan**		
Living with Japanese individuals		
Yes	110	22.1
No	387	77.9
Japanese level		
Good	244	49.1
So-so	253	50.9
Have confirmed family or friends in China		
Yes	161	32.4
No	336	67.6
Have suspected family or friends in China		
Yes	339	68.3
No	158	31.8

**Table 2 ijerph-18-04958-t002:** The prevalence of stress, anxiety, and depression.

Variables	Normal *n* (%)	Mild *n* (%)	Moderate *n* (%)	Severe *n* (%)	Extremely Severe *n* (%)
**Stress**	25 (5.03)	50 (10.06)	104 (20.93)	93 (18.71)	225 (45.27)
**Anxiety**	1 (0.2)	17 (3.42)	55 (11.07)	93 (18.71)	331 (66.60)
**Depression**	2 (0.4)	28 (5.63)	99 (19.92)	79 (15.90)	270 (54.33)

**Table 3 ijerph-18-04958-t003:** Association of disaster preparedness and social support with depression, anxiety, and stress.

	Stress			Anxiety			Depression		
	*B*	*SE*	*p*	*B*	*SE*	*p*	*B*	*SE*	*p*
**Disaster preparedness for resilience**	−0.095	0.016	0.000 **	−0.109	0.018	0.000 **	−0.119	0.019	0.000 **
**Social support**	−0.124	0.030	0.000 **	−0.172	0.033	0.000 **	−0.203	0.036	0.000 **
**Demographics**									
Age	−0.342	0.363	0.347	−1.034	0.399	0.010 *	0.040	0.425	0.926
Sex	−0.011	0.389	0.978	−1.478	0.427	0.826	−0.294	0.455	0.519
Education level	−0.929	0.426	0.030 *	−0.799	0.467	0.045 *	−0.432	0.498	0.050*
Occupation (ref = student)									
Medical staff	0.084	0.910	0.926	0.004	0.999	0.997	0.227	1.064	0.831
Educator	−0.636	0.929	0.494	−0.125	1.020	0.903	0.666	1.087	0.540
Other	−0.418	0.712	0.558	0.364	0.781	0.641	−0.323	0.832	0.698
Marriage	−0.710	0.480	0.139	−1.115	0.527	0.035 *	−0.181	0.561	0.747
Level of economic influence	0.826	0.384	0.032 *	0.842	0.421	0.046 *	0.972	0.449	0.031 *
**Variables related to COVID-19**									
Location in Japan	1.245	0.402	0.002 **	1.039	0.441	0.019 *	0.497	0.470	0.291
Time focused on COVID-19 news	0.684	0.264	0.010 *	0.258	0.290	0.374	0.372	0.309	0.229
Have COVID-19 symptoms	2.440	0.437	0.000 **	1.822	0.480	0.000 **	0.976	0.511	0.050 *
Had a PCR test	4.356	0.990	0.000 **	0.193	1.086	0.859	0.720	1.158	0.534
**Variables related to Chinese residents living in Japan**									
Living with Japanese individuals	−0.480	0.479	0.317	−1.352	0.526	0.317	−1.994	0.561	0.000
Japanese level	0.305	0.394	0.439	−0.574	0.433	0.439	0.712	0.461	0.123
Have confirmed family or friends in China	1.110	0.424	0.009 **	1.135	0.465	0.009 **	0.988	0.496	0.024 *
Have suspected family or friends in China	1.296	0.432	0.003 **	1.241	0.474	0.003 **	0.996	0.505	0.023 *

*B*: Correlation coefficient; *SE*: Std. Error. * *p* < 0.05. ** *p* < 0.01.

## Data Availability

The data presented in this study are available on reasonable request from the corresponding author.

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
