# Peer review of "Stress, Anxiety, and Depression for Chinese Residents in Japan during the COVID-19 Pandemic"

_ijerph, 2021, doi:10.3390/ijerph18094958_

Round 1
Reviewer 1 Report
This is a generally well-written manuscript detailing the results of an on-line survey of Chinese persons living in Japan during the COVID-19 crisis. The objective was to understand the impact of social support, preparedness, and other related factors on Depression, Anxiety, and Stress levels among respondents. The manuscript is certainly worthy of publication, but should first be modified to address the following:
- Although the English wording and phraseology are largely correct, there are several minor instances that indicate the paper should be submitted to editorial review by a native speaker of the English language. One example is the first sentence under section 2.2 which reads “All participants were voluntarily responded to the anonymous survey…” when it should say “All participants voluntarily responded to the anonymous survey…”
- The statement in the Introduction section that “Disaster preparedness is as helpful to foreigners during the pandemic as it is during natural disasters” should be referenced. I’m not certain that this is an established fact, but if it is, the reference citation would be nice to include.
- In the first paragraph of the Methods section, please indicate the exact ending “data collection date.”
- In describing the DASS-21, the statement is made that “The total scores for stress, anxiety, and depression, were classified as normal, mild, moderate, severe, and extremely severe.” However, the actual range of scores that would classify someone into these various categories is not specified. The ranges should be included.
- In the paragraph on covariates, please describe the “economic influence” variable in further detail. What does an influence of more or less than 50% actually mean? Is 50% half of the total family income? Please specify.
- Along these same lines, exactly what is meant by “time focused on COVID-19?” Is this the amount of personal education on the virus? The amount of news/media stories on which subjects focused? Or some other meaning. Please specify.
- Please be more explicit about the COVID symptom categories and the PCR test in section 3.1. Is the author saying that 98.5% did not have a PCR test, or he/she saying that 98.5% did not have a POSITVE PCR test? Is it the case that 68.2% of the whole sample of participants had families or associates in China AND 32.4% of the whole sample of participants had families or associates in China who had tested positive for COVID? Or is it the case that 68.2% had families or associates in China who tested positive and 32.4% of THAT SUBGROUP (i.e., people within the 68.2% subset) had tested positive for COVID? Also, when the author says 69.8% had no COVID symptoms, does this mean they had no symptoms at the time of the survey or that they had experienced no COVID symptoms during the entire epidemic?
- The authors should explicitly state the number of independent ANOVAs and independent regression statistics that were conducted. It looks like there were 27 ANOVAs and 3 regressions (which is quite a lot), but I’m not exactly sure about this.
- Related to the above, I’m a little concerned about the alpha inflation associated with so many independent statistical tests as well as the general strategy of conducting all of these ANOVA’s only to find that the results were basically the same as those from the regressions. I believe it would be more conservative to just go with the regressions and leave the ANOVA’s out altogether. Note the following: a) The regression equations showed that: “Among risk factors, participants with higher education levels were more likely to have lower stress (b = −0.93, p < 0.05), anxiety (b = −0.80, p < 0.05), and depression (b = −0.43, p < 0.05) symptoms. Those who reported a higher economic influence level were more likely to have higher stress (b = 0.83, p < 0.05), anxiety (b = 0.84, p < 0.05), and depression (b = 0.97, p < 0.05) symptoms. Those who had COVID-19 symptoms were more likely to have higher stress (b = 2.44, p < 0.01), anxiety (b 184 = 1.82, p < 0.01), and depression (b = 0.98, p < 0.05) symptoms. Those who had confirmed family or friends in China were more likely to have higher stress (b = 1.11, p < 0.01), anxiety (b = 1.14, p < 0.01), and depression (b = 0.99, p < 0.05) symptoms. Similarly, participants who had suspected family or friends in China were more likely to have higher stress (b = 188 1.30, p < 0.01), anxiety (b = 1.24, p < 0.01), and depression (b = 0.10, p < 0.05) symptoms.” AND b) The results from the ANOVAs were: “…higher education level was significantly related to lower stress 162 (P < 0.01), anxiety (P < 0.01), and depression (P < 0.01). Moreover, higher economic influence levels were also found to have a significant relationship with higher stress (P < 0.05), anxiety (P < 0.05), and depression (P < 0.05). For COVID-19 related factors, we found that COVID-19 symptoms were related to stress (P < 0.01), anxiety (P < 0.01), and depression (P < 0.05). The factors for Chinese living in Japan, those who have confirmed/suspected family or friend in China were significantly related to higher stress (P < 0.01), anxiety (P < 168 0.01), and depression (P < 0.01).” These two paragraphs are virtually identical, a fact that shows the ANOVAs were duplicative of the regression results. Why do we need these? I suggest deleting the ANOVAs altogether.
- Please carefully reword the second paragraph in the Discussion section. I believe it should read something like: “The results showed that 45.3% of the respondents indicated that they experienced stress during the pandemic, 66.6% reported heightened anxiety, and 54.3% indicated they experienced depression. The prevalence of these three symptoms in the present sample was comparatively higher than what was previously reported from a survey of 2000 local Japanese respondents. A total of 10.9% expressed pandemic-related anxiety and 17.3% reported depressive symptoms [38]. Studies of foreigners in other countries showed similar results. For example, 20% of foreign workers in Italy indicated they had experienced depression and 23% of the workers reported symptoms of anxiety during COVID-19—rates which were higher than those found among Italy’s native citizens [17]. It thus appears that foreigners as a group are more vulnerable to COVID-related increases in stress, anxiety, and depression than native-born citizens, a finding which agrees with earlier reports [39, 40].”
- Please consider rewording the second sentence in the limitations section to read: “The present results highlight the fact that foreigners living abroad, especially those who have families or friends affected by COVID-19 in their native countries, have a need for care that goes beyond that which is focused on citizens living in their native countries.”
Once these issues have been addressed. Publication of the paper should proceed. Thank you.
Reviewer 2 Report
Thank you for the opportunity to review the given manuscript. It is clearly presented and has some merits.
Some points need to be expanded & further described in the manuscript.
1.Sampling strategies (under the study design and participants): Your inclusion criteria were Chinese living in Japan who are adults. Please be specific by the definition "Chinese" are they recent immigrants to Japan? or who has lived in the country for a longer period of time? or are they children of Chinese migrants? What were your exact sampling strategies (please name them and describe the process of recruiting your participants). Did you reach out to certain online communities for the Chinese who are residing in Japan? Also, was the survey designed in the Chinese language only? If then, did you exclude Chinese whose Chinese language fluency was low? I am aware of the fact that Japan does not have the "immigration" system like other countries. Therefore the concept/definition of immigrants or migrants are different. But it is still an important information which will determine your target population, findings, and implications. As of now, this important information is missing. You need to address all of the points mentioned here to clearly present your study. If you were lacking in some aspects, you need to explicitly mention them in your limitations section.
2. Why study Chineses in Japan? : There have been numerous studies that were conducted and published in the literature of COVID-19 and individuals' mental health. What is the uniqueness of your study? Why did you study Chinese in Japan? What are their unique characteristics, challenges, and needs? This target population is not sufficiently introduced in your introduction section. Please have a separate paragraph to present this population (history of Chineses in Japan, their characteristics, numbers, etc). As of now, since this part is not covered well or in an intriguing way, it is difficult to be "hooked" to your study.
3. Ethical consideration: Please include the information on how the data was handled to protect respondents' confidentiality.
4. Discussion: Please include implications (practice or policy) for each of your findings. Your discussion section as of now explains your findings but doesn't expand further. You have these interesting and important findings, and you need to provide the "so what?" part.
5. Overall: why this study is important, what's novel contribution should be highlighted more (from abstract to introduction to conclusion).
Round 2
Reviewer 2 Report
Thank you for submitting the revised manuscript. It is much clearer and improved. I think that manuscript is ready to be published and will make a contribution to the literature.